# Verb Physics: Relative Physical Knowledge of Actions and Objects

## Abstract

Learning commonsense knowledge from natural language text is nontrivial due to *reporting bias*: people rarely state the obvious, e.g., "my house is *bigger* than me". However, while rarely stated explicitly, this trivial everyday knowledge does influence the way people talk about the world, which provides indirect clues to reason about the world. For example, a statement like "John *entered* his house" implies that his house is *bigger* than John.

In this paper, we present an approach to infer relative physical knowledge of actions and objects along six dimensions (e.g., size, weight, and strength) from unstructured natural language text. We frame knowledge acquisition as joint inference over two closely related problems: learning (1) relative physical knowledge of object pairs and (2) physical implications of actions when applied to those object pairs. Empirical results demonstrate that it is possible to extract knowledge of actions and objects from language and that joint inference over different knowledge types improves performance.

## 1 Introduction

Reading and reasoning about natural language text often requires trivial knowledge about everyday physical actions and objects. For example, given a sentence *"Martin could fit the trophy into the suitcase"*, we can trivially infer that the trophy must be smaller than the suitcase, even though it's not stated explicitly. This reasoning requires knowledge about the action *"fit"*, in particular, typical preconditions that need to be satisfied in order to perform the action. In addition, reasoning about

**Natural language clues**

*"She barged into the stable."*

**Relative physical knowledge about objects**

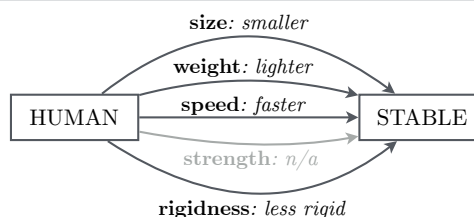

**Physical implications of actions**

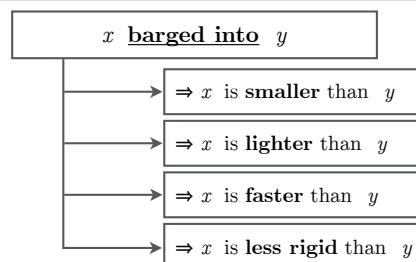

Figure 1: An overview of our approach. A verb's usage in language (top) implies physical relations between objects it takes as arguments. This allows us to reason about properties of specific objects (middle), as well as the knowledge implied by the verb itself (bottom).

the applicability of various physical actions in a given situation often requires background knowledge about objects in the world, for example, that people are usually *smaller* than houses, that cars generally move *faster* than humans walk, or that a brick probably is *heavier* than a feather.

In fact, the potential use of such knowledge about everyday actions and objects can go beyond language understanding and reasoning. Many open challenges in computer vision and robotics may also benefit from such knowledge, as shown

in recent work that requires visual reasoning and entailment (Izadinia et al., 2015; Zhu et al., 2014). Ideally, an AI system should acquire such knowledge through direct physical interactions with the world. However, such a physically interactive system does not seem feasible in a foreseeable future.

In this paper, we present an approach to acquire trivial physical knowledge from unstructured natural language text as an alternative knowledge source. In particular, we focus on acquiring relative physical knowledge of actions and objects organized along five dimensions: size, weight, strength, rigidness, and speed. Figure 1 illustrates example knowledge of (1) relative physical relations on object pairs and (2) physical implications of actions when applied to those object pairs.

While natural language text is a rich source to obtain broad knowledge about the world, compiling trivial commonsense knowledge from unstructured text is a nontrivial feat. The central challenge lies in *reporting bias*: people rarely states the obvious (Gordon and Van Durme, 2013; Sorower et al., 2011), since it goes against Grice's conversational maxim on the quantity (Grice, 1975) of information.

In this work, we demonstrate that it is possible to overcome reporting bias and still extract the unspoken knowledge from language. The key insight is this: there is consistency in the way people describe how they interact with the world, which provides vital clues to reverse engineer the common knowledge shared among people. More concretely, we frame knowledge acquisition as joint inference over two closely related puzzles: inferring relative physical knowledge about object pairs while simultaneously reasoning about physical implications of actions.

Importantly, four of five dimensions of knowledge in our study — weight, strength, rigidness, speed — are either not visual or not easily recognizable from image recognition using currently available computer vision techniques. Thus, our work provides unique values to complement recent attempts to acquire commonsense knowledge from web images (Izadinia et al., 2015; Bagherinezhad et al., 2016; Sadeghi et al., 2015).

In sum, our contributions are threefold:

- We introduce a new task of commonsense knowledge extraction from language, focusing on physical implications of actions and relative physical relations among objects, organized along five dimensions.
- We propose a model that can infer relations over grounded object pairs together with first order relations implied by physical verbs.
- We develop a new dataset VERBPHYSICS that compiles crowd-sourced knowledge of actions and objects.[1]

The rest of the paper is organized as follows. We first provide the formal definition of knowledge we aim to learn in Section 2. We then describe our data collection in Section 3 and present our inference model in Section 4. Empirical results and related work follow in Sections 5 and Section 6. We conclude at Section 7.

## 2 Representation of Relative Physical Knowledge

### 2.1 Knowledge Dimensions

We consider five dimensions of relative physical knowledge in this work: *size, weight, strength, rigidness,* and *speed.* "Strength" in our work refers to the physical durability of an object (e.g., "diamond" is stronger than "glass"), while "rigidness" refers to the physical flexibility of an object (e.g., "glass" is more rigid than a "wire"). When considered in verb implications, *size, weight, strength, rigidness* generally concerns pre-conditions of the action, while *speed* concerns the post-condition of the action.

### 2.2 Relative physical knowledge

Let us first consider the problem of representing relative physical knowledge between two objects. We can write a single piece of knowledge like "A person is larger than a basketball" as

$$\texttt{person} >^{\text{size}} \texttt{basketball}$$

Any propositional statement can have exceptions and counterexamples. Moreover, we need to cope with uncertainties involved in knowledge acquisition. Therefore, we assume each knowledge piece is associated with a probability distribution. More formally, given objects $x$ and $y$, we define a random variable $O_{x,y}^a$ whose range is $\{\boxed{>}, \boxed{<}, \boxed{\simeq}\}$ with respect to a knowledge dimension $a \in \{\text{SIZE}, \text{WEIGHT}, \text{STRENGTH}, \text{RIGIDNESS}, \text{SPEED}\}$ so that:

$$P(O_{x,y}^a = r), r \in \{\boxed{>}, \boxed{<}, \boxed{\simeq}\}.$$

---

[1]Will be publicly shared at `Anonymized.URL`.

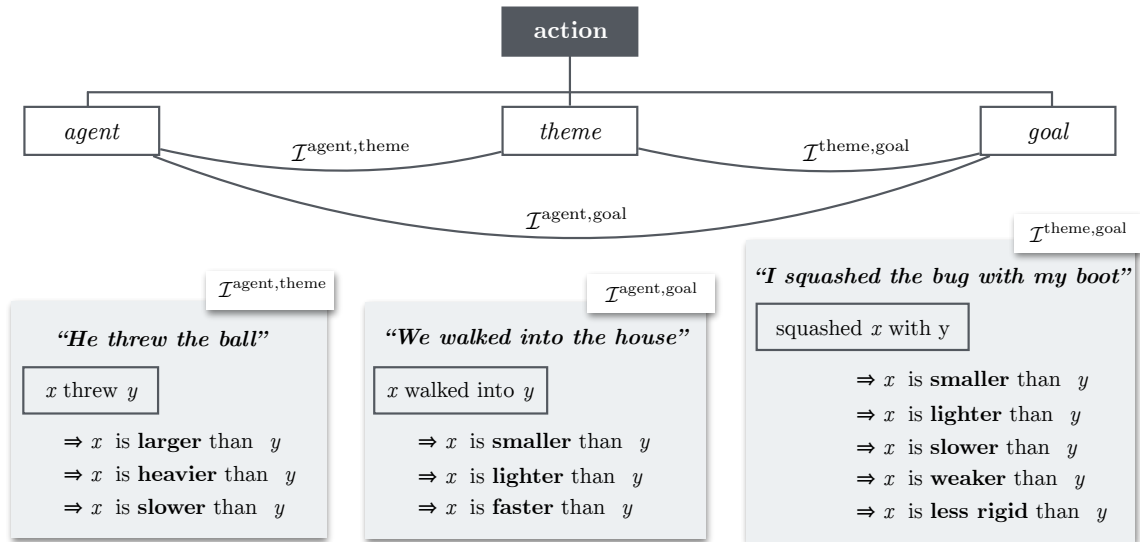

Figure 2: Example physical implications represented as frame relations between a pair of arguments.

This immediately provides two simple properties:

$$P(O_{x,y} = \boxed{>}) = P(O_{y,x} = \boxed{<})$$
$$P(O_{x,x} = \boxed{\approx}) = 1$$

### 2.3 Physical Implications of Verbs

Next we consider representing relative physical implications of actions applied over two objects. For example, consider an action frame "$x$ threw $y$." In general, following implications are likely to be true:

$$\text{"}x \text{ threw } y\text{"} \implies x >^{\text{size}} y$$
$$\text{"}x \text{ threw } y\text{"} \implies x >^{\text{weight}} y$$

Again, in order to cope with exceptions and uncertainties, we assume a probability distribution associated with each implication. More formally, we define a random variable $F_v^a$ to denote the implication of the action verb $v$ when applied over its arguments $x$ and $y$ with respect to a knowledge dimension $a$ so that:

$$P(F_{\text{threw}}^{\text{size}} = \boxed{>}) := P(\text{"}x \text{ threw } y\text{"} \Rightarrow x >^{\text{size}} y)$$
$$P(F_{\text{threw}}^{\text{wgt}} = \boxed{>}) := P(\text{"}x \text{ threw } y\text{"} \Rightarrow x >^{\text{wgt}} y)$$

where the range of $F_{threw}^{size}$ is $\{\boxed{>}, \boxed{<}, \boxed{\approx}\}$. Intuitively, $F_{threw}^{size}$ represents the likely first order relation implied by "throw" over ungrounded (i.e., variable) object pairs.

Above definition assumes that there is only a single implication relation for any given verb with respect to a specific knowledge dimension. This is generally not true, since a verb, especially a common action verb, can often invoke a number of different frames according to frame semantics (Fillmore, 1976). Thus, given a number of different frame relations $v_1...v_T$ associated with a verb $v$, we define random variables $F$ with respect to a specific frame relation $v_t$, i.e., $F_{v_t}^a$. We use this notation going forward.

**Frame Perspective on Verb Implications:** Figure 2 illustrates the frame centric view on physical implication knowledge we aim to learn. Importantly, the key insight of our work is inspired by Fillmore's original manuscript on frame semantics (Fillmore, 1976). Fillmore has argued that "frames"—the contexts in which utterances are situated—should be considered as a third primitive of describing a language, along with a grammar and lexicon. While existing frame annotations such as FrameNet (Baker et al., 1998), PropBank (Palmer et al., 2005), and VerbNet (Schuler, 2005) provide a rich frame knowledge associated with a predicate, none of them provide the exact kind of physical implications we consider in our paper. Thus, our work can potentially contribute to these resources by investigating new approaches to automatically recover richer frame knowledge from language.

## 3 Data and Crowd-sourced Knowledge

**Action Verbs:** We pick 50 classes of Levin verbs from both "alternation classes" and "verb

classes" (Levin, 1993), which corresponds to about 1100 unique verbs. We sort this list by frequency of occurrence in our frame patterns in the Google Syntax Ngrams corpus (Goldberg and Orwant, 2013) and pick the top 100 verbs.

**Action Frames:** Figure 2 illustrates examples of action frame relations. Because we consider implications over pairwise argument relations for each frame, there are sometimes multiple frame relations we consider for a single frame. To enumerate action frame relations for each verb, we use syntactic patterns based on dependency parse by extracting the core components (subject, verb, direct object, prepositional object) of an action, then map the subject to an agent, the direct object to a theme, and the prepositional object to a goal.[2] For those frames that involve an argument in a prepositional phrase, we create a separate frame for each preposition based on the statistics observed in the Google Syntax Ngram corpus.

Because the syntax ngram corpus provides only tree snippets without context, this way of enumerating potential frame patterns tend to over-generate. Thus we refine our prepositions for each frame by taking either the intersection or union with the top 5 Google Surface Ngrams (Michel et al., 2011), depending on whether the frame was under- or over-generating. We also add an additional crowdsourcing step where we ask crowd workers to judge whether a frame pattern with a particular verb and preposition could plausibly be found in a sentence. This process results in 813 frame templates, an average of 8.13 per verb.

**Object Pairs:** To provide a source of ground truth relations between objects, we select the object pairs that occur in the 813 frame templates with positive pointwise mutual information (PMI) across the Google Syntax Ngram corpus. After replacing a small set of "human" nouns with a generic HUMAN object, filtering out nouns labeled as abstract by WordNet (Miller, 1995), and distilling all surface forms to their lemmas (also with WordNet), the result is 3656 object pairs.

### 3.1 Crowdsourcing Knowledge

We collect human judgements of the frame knowledge implications to use as a small set of seed

---

[2]Future research could use an SRL parser instead. We use dependency parse to benefit from the Google Syntax Ngram dataset that provides language statistics over an extremely large corpus, which does not exist for SRL.

| Data collected | | |
|---|---|---|
| | **Total** | **Seed / dev / test** |
| Verbs | 100 | 5 / 45 / 50 |
| Frames | 813 | 65 / 333 / 415 |
| Object pairs$_{5\%}$ | 3656 | 183 / 1645 / 1828 |
| Object pairs$_{20\%}$ | " | 733 / 1096 / 1828 |

| Per attribute frame statistics | | | | |
|---|---|---|---|---|
| | *Agreement* | | *Counts* | |
| | **2/3** | **3/3** | **Verbs** | **Frames** |
| size | 0.91 | 0.41 | 96 | 615 |
| weight | 0.90 | 0.33 | 97 | 562 |
| strength | 0.88 | 0.25 | 95 | 465 |
| rigidness | 0.87 | 0.26 | 89 | 432 |
| speed | 0.93 | 0.36 | 88 | 420 |

| Per attribute object pair statistics | | | | |
|---|---|---|---|---|
| | *Agreement* | | *Counts* | |
| | **2/3** | **3/3** | **Distinct objs** | **Pairs** |
| size | 0.95 | 0.59 | 210 | 2552 |
| weight | 0.95 | 0.56 | 212 | 2586 |
| strength | 0.92 | 0.43 | 208 | 2335 |
| rigidness | 0.91 | 0.39 | 212 | 2355 |
| speed | 0.90 | 0.38 | 209 | 2184 |

Table 1: Statistics on Crowd-sourced Knowledge.

knowledge (5%), a development set (45%), and a test set (50%). Crowd workers are given with a frame template such as "**x** threw **y**," and then asked to list a few plausible objects (including people and animals) for the missing slots (e.g., **x** and **y**).[3] We then ask them to rate general relationship that the arguments of the frame exhibit with respect to the knowledge dimension we choose (size, weight, etc.). For each knowledge dimension, or attribute, $a$, workers select an answer from (1) $\mathbf{x} >^a \mathbf{y}$, (2) $\mathbf{x} <^a \mathbf{y}$, (3) $\mathbf{x} \simeq^a \mathbf{y}$, or (4) no general relation.

We conduct a similar crowdsourcing step for the set of object pairs. We ask crowd workers to compare each of the 3656 object pairs along the five knowledge dimensions we consider, selecting an answer from the same options above as with frames. We reserve 50% of the data as a test set, and split the remainder up either 5% / 45% or 20% / 30% (seed / development) to investigate the effects of different seed knowledge sizes on the model.

Statistics for the dataset are provided in Table 1. About 90% of the frames as well as object pairs had 2/3 agreement between workers. After removing frame/attribute combinations and object pairs that received less than 2/3 agreement, or were selected by at least 2/3 workers to have no relation,

---

[3]This step is to prime them for thinking about the particular template; we do not use the objects they provided.

we end up with roughly 400–600 usable frames and 2100–2500 usable object pairs per attribute.

## 4 Model

We model knowledge acquisition as probabilistic inference using a factor graph. As shown in Figure 3, the graph consists of multiple substrates (page-wide boxes) corresponding to different knowledge dimensions (shown only three of them —strength, size, weight—for brevity). Each substrate consists of two types of sub-graphs: verb subgraphs and object subgraphs, which are connected through factors that quantify action–object compatibilities. Connecting across substrates are factors that model inter-dependencies across different knowledge dimensions. In what follows, we describe each graph component.

### 4.1 Nodes

In the factor graph, we have two types of nodes in order to capture both classes of knowledge. The object first type of nodes are object pair nodes. Each object pair node is a random variable $O_{x,y}^a$ which captures the relative strength of attribute $a$ between objects $x$ and $y$.

The second type of nodes are frame nodes. Each frame node is a random variable $F_{v_t}^a$. This corresponds to the verb $v$ used in a particular type of frame $t$, and captures the implied knowledge the frame $v_t$ holds along an attribute $a$.

All random variables take on the values $\{\boxed{>}, \boxed{<}, \boxed{\simeq}\}$. For an object pair node $O_{x,y}^a$, the value represents the belief about the relation between $x$ and $y$ along attribute $a$. For a frame node $F_{v_t}^a$, the value represents the belief about the relation along attribute $a$ between *any* two objects that might be used in the frame $v_t$.

We denote the sets of all object pair and frame random variables $\mathcal{O}$ and $\mathcal{F}$, respectively.

### 4.2 Action–Object Compatibility

The key aspect of our work is to reason about two types of knowledge simultaneously: relative knowledge on grounded object pairs, and implications of actions related to those objects. Thus we connect the verb subgraphs and object subgraphs through selectional preference factors $\psi_s$ between two such nodes $O_{x,y}^a$ and $F_{v_t}^a$ if we find evidence from text that suggests objects $x$ and $y$ are used in the frame $v_t$. These factors encourage both random variables to agree on the same value.

As an example, consider a node $O_{p,b}^{size}$ which represents the relative size of a person and a basketball, and a node $F_{threw_{dobj}}^{size}$ which represents the relative size implied by an *"x threw y"* frame. If we find significant evidence in text that *"[person] threw [basketball]"* occurs, we would add a selectional preference factor to connect $O_{p,b}^{size}$ with $F_{threw_{dobj}}^{size}$ and encourage them towards the same value. This means that if it is discovered that people are larger than basketballs ($\boxed{>}$), then we would expect the frame *"x threw y"* to entail $x >^{size} y$ (also $\boxed{>}$).

### 4.3 Semantic Similarities

Some frames have relatively sparse text evidences to support their corresponding knowledge acquisition. Thus, we include several types of factors based on semantic similarities as described below.

**Cross-Verb Frame Similarity** We add a group of factors $\psi_v$ between two verbs $v$ and $u$ (to connect a specific frame of $v$ with a corresponding frame of $u$) based on the verb-level similarities.

**Within-Verb Frame Similarity** Within each verb $v$, which consists of a set of frame relations $v_1, ... v_T$, we also include frame-level similarity factors $\psi_f$ between $v_i$ and $v_j$. This gives us more evidence over a broader range of frames when textual evidence might be sparse.

**Object Similarity** As with verbs, we add factors $\psi_o$ that encourage similar pairs of objects to take the same value. Given that each node represents a pair of objects, finding that $x$ and $y$ are similar yields two main cases in how to add factors (aside from the trivial case where the variable $O_{x,y}^a$ is given a unary factor to encourage the value $\boxed{\simeq}$).

1. If nodes $O_{x,z}$ and $O_{y,z}$ exist, we expect objects $x$ and $y$ to both have a similar relation to $z$. We add a factor that encourages $O_{x,z}$ and $O_{y,z}$ to take the same value. The same is true if nodes $O_{z,x}$ and $O_{z,y}$ exist.

2. On the other hand, if nodes $O_{x,z}$ and $O_{z,y}$ exist, we expect these two nodes to reach the opposite decision. In this case, we add a factor that encourages one node to take the value $\boxed{>}$ if the other prefers the value $\boxed{<}$, and vice versa. (For the case of $\boxed{\simeq}$, if one prefers that value, then both should.)

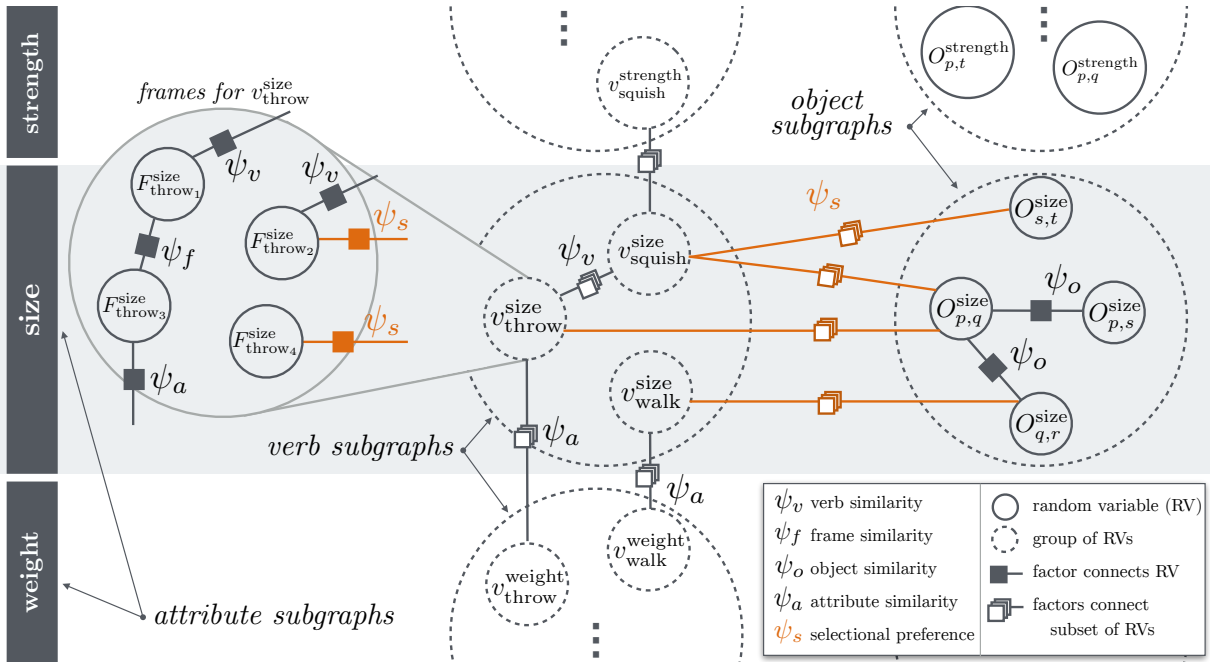

Figure 3: High level view of the factor graph model. Performance on both learning relative knowledge about objects (right), as well as entailed knowledge from verbs (center) via realized frames (left), is improved by modeling their interplay (orange). Unary seed ($\psi_{seed}$) and embedding ($\psi_{emb}$) factors are omitted for clarity.

### 4.4 Cross-Knowledge Correlation

Some knowledge dimensions, such as size and weight, have a significant correlation in their implied relations. For two such attributes $a$ and $b$, if the same frame $F_{v_i}^a$ and $F_{v_i}^b$ exists in both graphs, we add a factor $\psi_a$ between them to push them towards taking the same value.

### 4.5 Seed Knowledge

In order to kick off learning, we provide a small set of seed knowledge among the random variables in $\{\mathcal{O}, \mathcal{F}\}$ with seed factors $\psi_{seed}$. These unary seed factors push the belief for its associated random variable strongly towards the seed label.

### 4.6 Potential Functions

**Unary Factors** For all frame and object pair random variables in the training set, we train a maximum entropy classifier to predict the value of the variable. We then use the probabilities of the classifier as potentials for seed factors given to all random variables in their class (frame or object pair). Each log-linear classifier is trained separately per attribute on a featurized vector of the variable:

$$P(r|X^a) \propto e^{w_a \cdot f(X^a)}$$

The feature function is defined differently according to the node type:

$$f(O_{p,q}^a) := \langle g(p), g(q) \rangle$$
$$f(F_{v_t}^a) := \langle h(t), g(v), g(t) \rangle$$

Here $g(x)$ is the GloVe word embedding for the word $x$ ($t$ is the frame relation's preposition, and $g(t)$ is simply set to the zero vector if there is no preposition) and $h(t)$ is a one-hot vector of the frame relation type. We use GloVe vectors of 100 dimensions for verbs and 50 dimensions for objects and prepositions (the dimensions picked based on development set).

**Binary Factors** In the case of all other factors, we use a "soft 1" agreement matrix with strong signal down the diagonals:

$$\begin{bmatrix} & > & \simeq & < \\ > & \mathbf{0.7} & 0.1 & 0.2 \\ \simeq & 0.15 & \mathbf{0.7} & 0.15 \\ < & 0.2 & 0.1 & \mathbf{0.7} \end{bmatrix}$$

### 4.7 Inference

After our full graph is constructed, we use belief propagation to infer the assignments of frames and object pairs not in our training data. Each message $\mu$ is a vector $\mathbf{x}$ with probabilities for each value

$x \in \{\boxed{>}, \boxed{<}, \boxed{\simeq}\}$. A message passed from a random variable $v$ to a neighboring factor $f$ is the product of the messages from its other neighboring factors:

$$\mu_{v \to f}(x) \propto \prod_{f' \in N(v) \setminus \{f\}} \mu_{f' \to v}(x)$$

A message passed from a factor $f$ with potential $\psi$ to a random variable $v$ is a marginalized belief about the value of $v$ of the other neighboring random variables combined with its potential:

$$\mu_{f \to v}(x) \propto \sum_{\mathbf{x}' \in \mathbf{x}'_{\setminus x}} \psi(\mathbf{x}') \prod_{v' \in N(f) \setminus \{v\}} \mu_{v' \to f}(x'_{v'})$$

## 5 Experimental Results

**Factor Graph Construction** We first need to pick a set of frames and objects to determine our set of random variables. The frames are simply the subset of the frames that were crowdsourced in the given configuration (e.g., seed + dev), with "soft 1" unary seed factors (the gold label indexed row of the binary factor matrix) given only to those in the seed set. The same selection criteria and seed factors are applied to the crowdsourced object pairs.

For lexical similarity factors ($\psi_v$, $\psi_o$), we pick connections based on the cosine similarity scores of GloVe vectors (Pennington et al., 2014) thresholded above a value (0.4) chosen based on development set performance. Attribute similarity factors ($\psi_a$) are chosen based on sets of frames that reach largely the same decisions on the seed data (95%). Frame similarity factors ($\psi_f$) are added to pairs of frames with linguistically similar constructions. (Both attribute and frame similarity factors did not help on development set performance and thus were omitted from the final models; their effects are shown in the ablations in Table 3.)

Finally, selectional preference factors ($\psi_s$) are picked by using positive pointwise mutual information (PMI) between the frames and the object pairs' occurrences in the Google Syntax Ngram corpus.

**Baselines** Baselines include making a RANDOM choice, picking between $\boxed{>}$, $\boxed{<}$, and $\boxed{\simeq}$), picking the MAJORITY label, and a maximum entropy classifier based on the embedding representation (EMB-MAXENT).

### 5.1 Inferring Knowledge on Actions

Our first experiment is to predict knowledge implied by new frames. We experiment with two different seed knowledge sets: OUR MODEL (A) is based on more skimp seed knowledge, taking only 5% of the object pair data as seed. In contrast, OUR MODEL (B) is based on 20% of the object pairs as seed knowledge.

The full results for the baseline methods and our model are given in Table 4.7. Though the speed attribute has a skewed label distribution, giving the majority baseline high performance, our model outperforms the baselines on other attributes as well as overall.

**Metaphorical Language:** While our frame patterns are intended to capture action verbs, our templates also match senses of those verbs that can be used with abstract or metaphorical arguments, rather than directly physical ones. One example from the development set is "$x$ contained $y$." While $x$ and $y$ can be real objects, more abstract senses of "contained" could involve $y$ as a "forest fire" or even a "revolution." In these instances, $x >^{\text{size}} y$ is plausible as an abstract notion: if some entity can contain a revolution, we might think that entity as "larger" or "stronger" than the revolution.

### 5.2 Inferring Knowledge on Objects

Our second experiment is to predict the correct relations of new object pairs. Results for this task are also given in Table 4.7.

## 6 Related work

Several works straddle the gap between IE, knowledge base completion, and learning commonsense knowledge from text. Earlier works in these areas use large amounts of text to try to extract general statements like "A THING CAN BE READABLE" (Gordon et al., 2010) and frequencies of events (Gordon and Schubert, 2012). Our work focuses on specific domains of knowledge rather than general statements or occurrence statistics, and develops a frame- centric approach to circumvent reporting bias. Other work uses a knowledge base and scores unseen tuples based on similarity to existing ones (Angeli and Manning, 2013; Li et al., 2016). Relatedly, previous work uses natural language inference to infer new facts from a dataset of commonsense facts that can be ex-

| Algorithm | Development | | | | | | Test | | | | | |
|---|---|---|---|---|---|---|---|---|---|---|---|---|
| | size | weight | stren | rigid | speed | *overall* | size | weight | stren | rigid | speed | *overall* |
| RANDOM | 0.33 | 0.33 | 0.33 | 0.33 | 0.33 | 0.33 | 0.33 | 0.33 | 0.33 | 0.33 | 0.33 | 0.33 |
| MAJORITY | 0.38 | 0.41 | 0.42 | 0.18 | **0.83** | 0.43 | 0.35 | 0.35 | 0.43 | 0.20 | 0.88 | 0.44 |
| EMB-MAXENT | 0.62 | 0.64 | 0.60 | **0.83** | **0.83** | 0.69 | 0.55 | 0.55 | 0.59 | 0.79 | 0.88 | 0.66 |
| OUR MODEL (A) | 0.71 | 0.63 | 0.61 | 0.82 | **0.83** | 0.71 | 0.55 | 0.55 | 0.55 | 0.79 | **0.89** | 0.65 |
| OUR MODEL (B) | **0.74** | **0.69** | **0.67** | 0.82 | 0.78 | **0.74** | **0.76** | 0.59 | **0.66** | **0.80** | 0.87 | **0.73** |

| Algorithm | Development | | | | | | Test | | | | | |
|---|---|---|---|---|---|---|---|---|---|---|---|---|
| | size | weight | stren | rigid | speed | *overall* | size | weight | stren | rigid | speed | *overall* |
| RANDOM | 0.33 | 0.33 | 0.33 | 0.33 | 0.33 | 0.33 | 0.33 | 0.33 | 0.33 | 0.33 | 0.33 | 0.33 |
| MAJORITY | 0.50 | 0.54 | 0.51 | 0.50 | 0.53 | 0.51 | 0.51 | 0.55 | 0.52 | 0.49 | 0.50 | 0.51 |
| EMB-MAXENT | 0.68 | 0.66 | 0.64 | 0.67 | 0.65 | 0.66 | **0.71** | 0.67 | 0.64 | 0.65 | **0.63** | **0.66** |
| OUR MODEL (A) | **0.74** | **0.69** | **0.67** | **0.68** | **0.66** | **0.69** | 0.68 | **0.70** | **0.66** | **0.66** | 0.60 | **0.66** |

Table 2: Accuracy of baselines and our model on both tasks. Top: frame prediction task, OUR MODEL (A) uses 5% object pairs as seed, OUR MODEL (B) uses 20% object pairs as seed. Bottom: object pair prediction task (5% object pairs as seed).

| Ablated (or added) component | Accuracy |
|---|---|
| – Verb similarity | 0.69 |
| + Frame similarity | 0.62 |
| – Action-object compatibility | 0.62 |
| – Object similarity | 0.70 |
| + Attribute similarity | 0.62 |
| – Frame embeddings | 0.63 |
| – Frame seeds | 0.62 |
| – Object embeddings | 0.62 |
| – Object seeds | 0.62 |
| OUR MODEL (A) | **0.71** |

Table 3: Ablation results on *size* attribute for the frame prediction task on the development dataset for OUR MODEL (A) (5% of the object pairs as seed data). Our final model (OUR MODEL (B), using 20% object pairs as seed data), employs only unary embeddings and selctional preference factors.

tracted from unstructured text (Angeli and Manning, 2014). While our work also focuses on commonsense knowledge, we attempt to directly learn a small number of specific types of knowledge from text without reasoning from an existing database or dataset of facts.

A handful of works have attempted to learn the types of knowledge we address in this work. One recent work tried to directly predict several binary attributes (such "is large" and "is yellow") from on off-the-shelf word embeddings, noting that accuracy was very low (Rubinstein et al., 2015). Another line of work addressed grounding verbs in the context of robotic tasks. One paper in this line acquires verb meanings by observing state changes in the environment (She and Chai, 2016). While they have data on 165 verb frames, they only have enough to report statistical signifi-

cance on four of them. Another work in this line does a deep investigation of eleven verbs, modeling their physical effect via annotated images along eighteen attributes (Gao et al., 2016). These works are encouraging investigations into multimodal groundings of a small set of verbs. Our work instead grounds into a fixed set of attributes but leverages language on a broader scale to learn about more verbs in more diverse set of frames.

# 7 Conclusion

We presented a novel take on verb-centric frame semantics to learn implied physical knowledge latent in verbs. We showed that by modeling changes in physical attributes entailed by verbs together with objects that exhibit these properties, we are able to better solve both tasks. Experiments on our novel dataset confirm that a model which takes advantage of physical relations as they arise from verbs and between objects outperforms baselines lacking such information. Our dataset and code will be made publicly available upon publication.

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
