# Peer review of "Verb Physics: Relative Physical Knowledge of Actions and Objects"

_ACL 2017 — decision unknown_

[Official Review · Reviewer 1 · rating 3 · confidence 4]
soundness 4 · originality 4 · clarity 2 · impact 3 · substance 3 · appropriateness 5 · meaningful comparison 4 · presentation format Poster

Thank you for the author response. It addresses some my concerns, though much
of it are promises ("we will...") -- necessarily so, given space constraints,
but then, this is precisely the problem: I would like to see the revision to
the paper to be able to check that the drawbacks have been fixed. The changes
needed are quite substantial, and the new experimental results that they
promise to include will not have undergone review if the paper is accepted at
this stage. I'm still not sure that we can simply leave it to the authors to
make the necessary changes without a further reviewing round. I upgrade my
score to a 3 to express this ambivalence (I do like the research in the paper,
but it's extremely messy in its presentation).

--------------

- Strengths:

The topic of the paper is very creative and the purpose of the research really
worthwhile: the paper aims at extracting common knowledge from text, overcoming
the well-known problem of reporting bias (the fact that people will not state
the obvious, such as the fact that a person is usually bigger than a ball), by
doing joint inference on information that is possible to extract from text.

- Weaknesses:

1) Many aspects of the approach need to be clarified (see detailed comments
below). What worries me the most is that I did not understand how the approach
makes knowledge about objects interact with knowledge about verbs such that it
allows us to overcome reporting bias. The paper gets very quickly into highly
technical details, without clearly explaining the overall approach and why it
is a good idea.

2) The experiments and the discussion need to be finished. In particular, there
is no discussion of the results of one of the two tasks tackled (lower half of
Table 2), and there is one obvious experiment missing: Variant B of the
authors' model gives much better results on the first task than Variant A, but
for the second task only Variant A is tested -- and indeed it doesn't improve
over the baseline. 

- General Discussion:

The paper needs quite a bit of work before it is ready for publication. 

- Detailed comments:

026 five dimensions, not six

Figure 1, caption: "implies physical relations": how do you know which physical
relations it implies?

Figure 1 and 113-114: what you are trying to do, it looks to me, is essentially
to extract lexical entailments (as defined in formal semantics; see e.g. Dowty
1991) for verbs. Could you please explicit link to that literature?

Dowty, David. "Thematic proto-roles and argument selection." Language (1991):
547-619.

135 around here you should explain the key insight of your approach: why and
how does doing joint inference over these two pieces of information help
overcome reporting bias?

141 "values" ==> "value"?

143 please also consider work on multimodal distributional semantics, here
and/or in the related work section. The
following two papers are particularly related to your goals:

Bruni, Elia, et al. "Distributional semantics in technicolor." Proceedings of
the 50th Annual Meeting of the Association for Computational Linguistics: Long
Papers-Volume 1. Association for Computational Linguistics, 2012.

Silberer, Carina, Vittorio Ferrari, and Mirella Lapata. "Models of Semantic
Representation with Visual Attributes." ACL (1). 2013.

146 please clarify that your contribution is the specific task and approach --
commonsense knowledge extraction from language is long-standing task.

152 it is not clear what "grounded" means at this point

Section 2.1: why these dimensions, and how did you choose them?

177 explain terms "pre-condition" and "post-condition", and how they are
relevant here

197-198 an example of the full distribution for an item (obtained by the model,
or crowd-sourced, or "ideal") would help.

Figure 2. I don't really see the "x is slower than y" part: it seems to me like
this is related to the distinction, in formal semantics, between stage-level
vs. individual-level
predicates: when a person throws a ball, the ball is faster than the person
(stage-level) but
it's not true in general that balls are faster than people (individual-level).
I guess this is related to the
pre-condition vs. post-condition issue. Please spell out the type of
information that you want to extract.

248 "Above definition": determiner missing

Section 3

"Action verbs": Which 50 classes do you pick, and you do you choose them? Are
the verbs that you pick all explicitly tagged as action verbs by Levin? 

306ff What are "action frames"? How do you pick them?

326 How do you know whether the frame is under- or over-generating?

Table 1: are the partitions made by frame, by verb, or how? That is, do you
reuse verbs or frames across partitions? Also, proportions are given for 2
cases (2/3 and 3/3 agreement), whereas counts are only given for one case;
which?

336 "with... PMI": something missing (threshold?)

371 did you do this partitions randomly?

376 "rate *the* general relationship"

378 "knowledge dimension we choose": ? (how do you choose which dimensions you
will annotate for each frame?)

Section 4

What is a factor graph? Please give enough background on factor graphs for a CL
audience to be able to follow your approach. What are substrates, and what is
the role of factors? How is the factor graph different from a standard graph?
More generally, at the beginning of section 4 you should give a higher level
description of how your model works and why it is a good idea.

420 "both classes of knowledge": antecedent missing.

421 "object first type"

445 so far you have been only talking about object pairs and verbs, and
suddenly selectional preference factors pop in. They seem to be a crucial part
of your model -- introduce earlier? In any case, I didn't understand their
role.

461 "also"?

471 where do you get verb-level similarities from?

Figure 3: I find the figure totally unintelligible. Maybe if the text was
clearer it would be interpretable, but maybe you can think whether you can find
a way to convey your model a bit more intuitively. Also, make sure that it is
readable in black-and-white, as per ACL submission instructions.

598 define term "message" and its role in the factor graph.

621 why do you need a "soft 1" instead of a hard 1?

647ff you need to provide more details about the EMB-MAXENT classifier (how did
you train it, what was the input data, how was it encoded), and also explain
why it is an appropriate baseline.

654 "more skimp seed knowledge": ?

659 here and in 681, problem with table reference (should be Table 2). 

664ff I like the thought but I'm not sure the example is the right one: in what
sense is the entity larger than the revolution? Also, "larger" is not the same
as "stronger".

681 as mentioned above, you should discuss the results for the task of
inferring knowledge on objects, and also include results for model (B)
(incidentally, it would be better if you used the same terminology for the
model in Tables 1 and 2)

778 "latent in verbs": why don't you mention objects here?

781 "both tasks": antecedent missing

The references should be checked for format, e.g. Grice, Sorower et al
for capitalization, the verbnet reference for bibliographic details.

[Official Review · Reviewer 2 · rating 4 · confidence 4]
soundness 4 · originality 4 · clarity 4 · impact 3 · substance 4 · appropriateness 5 · meaningful comparison 4 · presentation format Oral Presentation

Summary: This paper aims to learn common sense relationships between object
categories (e.g comparative size, weight, strength, rigidness, and speed) from
unstructured text.  The key insight of the paper is to leverage the correlation
of action verbs to these comparative relations (e.g x throw y => x larger y).

Strengths:

- The paper proposes a novel method to address an important problem of mining
common sense attribute relations from text.

Weaknesses:

- I would have liked to see more examples of objects pairs, action verbs, and
predicted attribute relations.                          What are some interesting
action
verbs
and
corresponding attributes relations?  The paper also lacks analysis/discussion 
on what kind of mistakes their model makes.

- The number of object pairs (3656) in the dataset is very small.  How many
distinct object categories are there?  How scalable is this approach to larger
number of object pairs?

- It's a bit unclear how the frame similarity factors and attributes similarity
factors are selected.

General Discussion/Suggestions:

- The authors should discuss the following work and compare against mining
attributes/attribute distributions directly and then getting a comparative
measure.  What are the advantages offered by the proposed method compared to a
more direct approach?

Extraction and approximation of numerical attributes from the Web
Dmitry Davidov, Ari Rappoport
ACL 2010

Minor typos:

1. In the abstract (line 026), the authors mention 'six' dimensions, but in the
paper, there is only five.

2. line 248: Above --> The above

3. line 421: object first --> first

4. line 654: more skimp --> a smaller

5. line 729: selctional --> selectional

[Official Review · Reviewer 3 · rating 4 · confidence 4]
soundness 4 · originality 4 · clarity 5 · impact 3 · substance 4 · appropriateness 5 · meaningful comparison 4 · presentation format Oral Presentation

The paper studies an interesting problem of extracting relative physical
knowledge of actions and objects from unstructured text, by inference over a
factor graph that consists of two types of subgraphs---action graph and object
graph. The paper stems from the key insight---common knowledge about physical
world influences the way people talk, even though it is rarely explicitly
stated. 

- Strengths:

The paper tries to solve an interesting and challenging problem. The problem is
hard due to reporting bias, and the key insight/approach in the paper is
inspiring.

The model is innovative and clearly described. And the idea of handling text
sparsity with semantic similarity factors is also appropriate. 

The empirical evidence well supports the effectiveness of the model (compared
to other baselines). 

The paper is well-written, with informative visualization, except for some
minor errors like *six dimensions* in abstract but *five* everywhere else. 

- Weaknesses:

The benefits and drawbacks of model components are still somehow
under-discussed, and hard to tell with the limited quantitative results in the
paper. 

For example, is there any inherent discrepancy between *cross-verb frame
similarity*, *within-verb frame similarity* and *action-object compatibility*?
Frames of *A throw B* and *C thrown by D* share a verb primitive *throw*, so
should it infer C>D (by *within-verb*) if A>B is given? 
On the other side,
frames of *C thrown by D* and *E kicked by F* share the frame *XXX by*, so if
F>E is known, is D>C inferred? How does the current model deal with such
discrepancy?

The paper might be better if it has more qualitative analysis. And more
evidence also needs to be provided to gauge how difficult the task/dataset is.

For example, are the incorrectly-classified actions/objects also ambiguous for
human? On what types of actions/objects does the model tend to make mistakes?
Is the verb with more frame types usually harder than others for the model?

More interestingly, how are the mistakes made? Are they incorrectly enforced by
any
proposed *semantic similarity*?

I think more analysis on the model components and qualitative results may
inspire more general framework for this task. 

- General Discussion:

/* After author response */

After reading the response, I tend to keep my current rating and accept this
paper. The response well addresses my concerns. And I tend to believe that
necessary background and experimental analysis can be added given some
re-organization of the paper and one extra page, as it is not hard. 

/* Before author response */

I think this paper is in general solid and interesting. 
I tend to accept it, but it would be better if the questions above can be
answered.